# Enhanced Tensorial Self-representation Subspace Learning for Incomplete Multi-view Clustering

## ABSTRACT

Incomplete Multi-View Clustering (IMVC) is a promising topic in multimedia as it breaks the data completeness assumption. Most existing methods solve IMVC from the perspective of graph learning. In contrast, self-representation learning enjoys a superior ability to explore relationships among samples. However, only a few works have explored the potentiality of self-representation learning in IMVC. These self-representation methods infer missing entries from the perspective of whole samples, resulting in redundant information. In addition, designing an effective strategy to retain salient features while eliminating noise is rarely considered in IMVC. To tackle these issues, we propose a novel self-representation learning method with missing sample recovery and enhanced low-rank tensor regularization. Specifically, the missing samples are inferred by leveraging the local structure of each view, which is constructed from available samples at the feature level. Then an enhanced tensor norm, referred to as Logarithm-$p$ norm is devised, which can obtain an accurate cross-view description. Our proposed method achieves exact subspace representation in IMVC by leveraging high-order correlations and inferring missing information at the feature level. Extensive experiments on several widely used multi-view datasets demonstrate the effectiveness of the proposed method.

## CCS CONCEPTS

• **Computing methodologies** → **Cluster analysis**; • **Information systems** → **Clustering**.

## KEYWORDS

Incomplete multi-view clustering, self-representation learning, low-rank tensor learning

## 1 INTRODUCTION

Multi-view learning has become a popular direction of machine learning due to the excellent expressive capability of multi-view data. Due to the rich complementary information embedded in multi-view data, multi-view clustering methods have been well-developed. Compared to single-view clustering, multi-view clustering can exploit the consistency and complementary information across the different views, leading to superior performance.

*ACM MM, 2024, Melbourne, Australia*
© 2024 Copyright held by the owner/author(s). Publication rights licensed to ACM.
ACM ISBN 978-x-xxxx-xxxx-x/YY/MM
https://doi.org/10.1145/nnnnnnn.nnnnnnn

Roughly, current multi-view clustering approaches can be categorized into the following four types: matrix factorization clustering [11, 43], graph-based clustering [23, 24, 34, 41], deep multiview clustering [28, 52, 54], and multiview subspace clustering [5, 10, 32, 45, 57]. In particular, self-representation learning methods based on low-rank tensor regularization achieve commendable performance, where they mine complementary information from multiple views by tensor Singular Value Decomposition (t-SVD) based nuclear norm [8, 13, 51, 53].

Most of the existing methods are based on the completeness assumption, which denotes that samples in all views are observed. However, in real-world scenarios, collected instances are usually missing in certain views due to malfunctions in sensors or data corruption during transmission [12]. The absent views create a significant challenge for multi-view clustering where the consistent information across multiple views is difficult to learn. To tackle this limitation, numerous researchers have devoted themselves to IMVC, leading to the flourishing development of IMVC. As a foundational work in the field of IMVC, Partial Multi-view Clustering (PVC) [21] utilizes the Non-negative Matrix Factorization (NMF) to establish a latent subspace under incomplete conditions. Expanding from PVC, several impressive algorithms have emerged [17, 36, 59]. The work in [59] introduces a novel graph Laplacian term to bridge the connection of missing samples across multiple views. In [17] and [36], enhanced NMF, including weighted NMF and weighted semi-NMF are adopted to improve the ability to handle incomplete multi-view data. The works in [48, 55] reduce the impact of incomplete instances and capture consistent information by inferring the missing entries.

Apart from the above matrix factorization based methods, the graph learning framework is introduced into IMVC. Perturbation-oriented IMVC [42] transfers the missing problem from the perspective of instances to the graph and reduces the spectral perturbation risk among different views. Wen *et al.* [47] propose an adaptive graph completion strategy and unify the graph completion and consensus graph learning into one framework. Liang *et al.* [26] devise a graph fusion-based IMVC model that alleviates the impact of missing instances by sample-level auto weights. Liu *et al.* [27] infer the missing entries in partial graphs by graph propagation strategy. The work in [46] directly learns the clustering results from a consensus probability representation. Zhao *et al.* [60] introduce a heterogeneous-graph learning and embedding strategy to adopt high-order structures and recover the incomplete graph for each view.

Although graph-based approaches have achieved impressive performance, they still endure high computational complexity. In order to enhance efficiency, anchor technology is introduced. Nie *et al.* [6] and Wang *et al.* [40] leverage anchor graph for complete multi-view clustering. To expand anchor graph learning into IMVC, the following works have emerged. Anchor-based partial multi-view

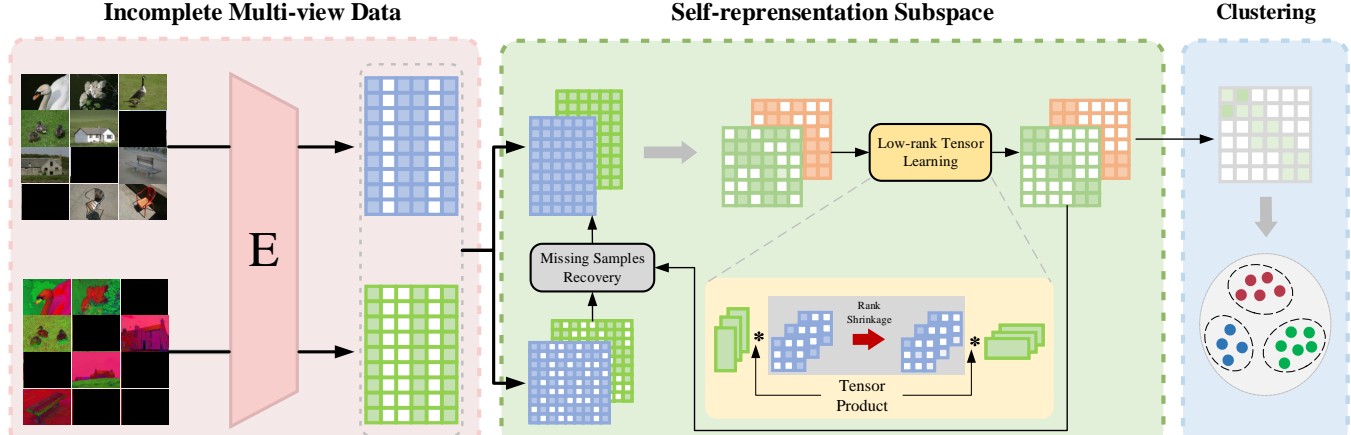

**Figure 1: The framework of the proposed ELRSSL. Two views are illustrated herein. From left to right, the first step is extracting features from incomplete multi-view data. The local structure corresponding to each view is calculated to infer the missing samples. Then, self-representation subspaces are learned, meanwhile, the low-rank tensor learning is jointly performed. Finally, the clustering results are obtained by applying spectral clustering.**

clustering [16] reconstructs similarity among instances by anchors. The work in [44] proposes a highly efficient multi-view clustering method based on a bipartite graph framework for incomplete data. Recently, a scalable IMVC method [50] has been proposed, which constructs the view-specific anchor graph for capturing complementary information and devises a novel structure alignment module to eliminate the misalignment of anchors.

The methods mentioned above have improved the performance of IMVC from various perspectives, most of them are based on matrix factorization or graph learning. Numerous multi-view clustering methods have empirically demonstrated that self-representation subspace learning and low-rank tensor constraint are beneficial for clustering [7, 13, 15, 18, 29, 39, 53]. For complete multi-view data, the work in [53] is the pioneer in adopting low-rank tensor in self-representation subspace learning. Motivated by [53], t-SVD based methods are well-developed [3, 35, 51]. To overcome the gap of self-representation approaches in IMVC task, missing data imputation and self-representation are unified into a framework [29]. Further, a tensor factorization term is adopted to preserve high-order correlation in subspaces [25]. However, the used strategy is to infer absent samples by summarising information from whole observed samples, resulting in redundant information. In addition, the reservations about salient features and the elimination of noise are neglected during exploring high-order correlation.

To tackle this issue, we propose an Enhanced Low-Rank tensor Self-representation Subspace Learning (ELRSSL) for IMVC. In detail, our ELRSSL recovers the missing instance by restricting similar features which ensures the inferred entries are meaningful, and index matrices are utilized to ensure the inferred entries belong to missing samples. Meanwhile, a low-rank tensor learning strategy is adopted to explore the high-order correlation across multiple views. In addition, the large singular values are considered to be important because they contain main feature information, while small singular values represent the noise embedded in data. Weighted Tensor Nuclear Norm (WTNN), which shrinks singular values by assigned

weights, is a popular technology for low-rank tensor learning [13, 15]. However, WTNN needs the a priori information about data to assign reasonable weights. For the proposed method, an enhanced nuclear norm is devised, which assigns weight to various singular values with a self-adaptive weight strategy. The overall framework is depicted in Figure 1. The main contribution of this paper is summarised as follows.

- We present ELRSSL, a self-representation approach for IMVC which enhances the expressive capability of subspace on incomplete data by exploring high-order correlation and missing information inferences at the feature level.
- An enhanced nuclear norm is devised, which assigns weight to different singular values. The scale of weights is determined by corresponding singular values. Thus, the important features across various views are retained while the noises embedded in multiple views are eliminated.
- To solve the proposed method, an alternating optimization algorithm is designed with a computational complexity analysis. The extensive experiments demonstrate the superiority of the proposed method.

## 2 RELATED WORK

### 2.1 Self-representation Clustering

Multi-view self-representation clustering relies on all instances to construct coefficient matrices from multiple views. Multi-view self-representation clustering can reflect the relationships among instances, leading to excellent clustering performance. According to [19, 57], the self-representation clustering for complete data can be formulated into

$$\min_{Z^v} \sum_{v=1}^{V} \mathcal{L}(X^v - X^v Z^v) + \mathcal{R}(Z^v), \quad (1)$$

where $Z^v$ is the subspace corresponding to $v$th view, the first term denotes the loss of subspace coefficient reconstruction, and the

second term is the regularity of $Z^v$. In literature, $\mathcal{L}$ and $\mathcal{R}$ can be replaced with various norms, e.g. $l_2$-norm, $l_1$-norm, $l_{2,1}$-norm, graph regularity, and nuclear norm, etc. Luo *et al.* [33] exploit consistency and specificity jointly for subspace. Zhang *et al.* [56] utilize naive low-rank regularity to learn the subspace structure. Moreover, Cai *et al.* [3] introduce a high-order manifold regularized term to capture the manifold information of data in subspace learning. However, this naive strategy neglects the impact of the missing instances without any completion term. To infer the missing information, our ELRSSL leverages a local structure at the feature level which is restricted to similar features and ensures the inferred features are meaningful.

## 2.2 Low-rank Tensor Learning

Low-rank tensor learning has been empirically proven to be effective in capturing high-order correlation [8, 13, 35, 51, 53]. The formula for low-rank tensor learning is given as

$$\min_{\mathcal{Z}} \mu rank(\mathcal{Z}) + \mathcal{L}(\mathcal{Z} - \mathcal{Y}), \tag{2}$$

where $\mathcal{Z}$ and $\mathcal{Y}$ denote the learned low-rank tensor and approximation tensor, respectively. The $rank()$ is the estimation of tensor rank which is usually replaced with various nuclear norms, e.g. TNN [53], weighted TNN [13] etc. Since the assumption that the large singular values are generally associated with some salient parts while the small singular values are the contrary, a natural idea is to shrink large singular values less to preserve the main information. Some remarkable approaches are developed, one popular strategy is the WTNN [13–15]. Besides, the nonconvex low-rank tensor approximation is widely used in various fields [8, 9]. Distinct from the above works, our proposed ELRSSL adopts an enhanced low-rank tensor regularity, enables the important information can be preserved adaptively.

## 2.3 Incomplete Multi-view Clustering

Given an incomplete dataset $\{X^v\}_{v=1}^V$ and indicator vector $\{e^v\}_{v=1}^V$, where $V$ is the number of views, each element of $e^v$ is defined as follows.

$$e_i^v = \begin{cases} 1, & \text{If } i\text{th instance is observed in } v\text{th view,} \\ 0, & \text{Otherwise.} \end{cases} \tag{3}$$

The goal of IMVC is to partition the instances under the incomplete condition. Based on the NMF, some simple but effective methods are devised [17, 21, 36, 59]. Beyond, graph-based methods [20, 22, 37] also achieve impressive performance, e.g. work in [38] learns consensus graph and clustering indicator simultaneously, Liu *et al.* [27] propose a graph propagation strategy to infer missing entries by graph propagation, Guo *et al.* [16] captures non-linear relations by kernel on computed anchor graph. Based on anchor technology, the work in [30] proposes an algorithm with linear time complexity and space complexity. Wen *et al.* [49] simultaneously consider the local information, the complementary information, and the discriminatory power of all views. Liu *et al.* [29] first introduce self-representation subspace clustering to IMVC. Unlike prior works, we step further by mining the high-order information while preserving the salient feature as well as ensuring that the inferred information is useful.

# 3 THE PROPOSED METHOD

Incomplete multi-view data presents a great challenge to clustering tasks. To implement self-representation subspace learning to IMVC, the missing information completion and consistent correlation learning are crucial. Therefore, we devise a novel self-representation method for IMVC as follows.

## 3.1 Methodology

To complete the missing instances, the local structure is leveraged. The completion matrix $E^v \in \mathbb{R}^{d_v \times n_v^m}$ and index matrix $W^v \in \mathbb{R}^{n_v^m \times n}$ corresponding to $v$th view are introduced, where $d_v$ and $n_v^m$ denote the dimensions of feature and number of missing instances. The index matrix $W^v$ is constructed by

$$W_{i,j}^v = \begin{cases} 1, & \text{If } i\text{th missing instance is } j\text{th instance in } v\text{th view,} \\ 0, & \text{Otherwise.} \end{cases} \tag{4}$$

The completion matrix $E^v$ is learned from the existing information by

$$\min_{E^v} \sum_{v=1}^V \sum_{i,j}^{d_v} \|E_{i,:}^v - Ej, :^v\|_2^2 G_{i,j}^v, \tag{5}$$

where $G^v \in \mathbb{R}^{d_v \times d_v}$ is a guidance matrix used to ensure the recovered missing samples come from similar features. $G^v$ is a similarity matrix constructed in the feature perspective.

$$G_{i,j}^v = \begin{cases} 1, & \text{If } i\text{th and } j\text{th features are} \\ & \text{mutually the } k \text{ closest neighbors,} \\ 0, & \text{Otherwise.} \end{cases} \tag{6}$$

The above term completes samples from the feature level while ensuring the recovered features are meaningful by restricting similar features [4]. According to the normalization of the Laplacian matrix, Formula (5) can be reformulated as

$$\min_{E^v} \sum_{v=1}^V Tr((E^v)^T L_g^v E^v), \tag{7}$$

where $L_g^v \in \mathbb{R}^{d_v \times d_v}$ is a Laplacian matrix computed by $L_g^v = D_g^v - G^v$. The $D_g^v$ which is calculated as $D_g^v = diag(\{\sum_j^{d_v} G_{i,j}^v\}_{i=1}^{d_v})$, is a degree matrix of $G^v$. Then, we learn the complete feature view with the following model.

$$\min_{E^v, \hat{X}^v} \sum_{v=1}^V (\|X^v + E^v W^v - \hat{X}^v\|_F^2 + \alpha Tr((E^v)^T L_g^v E^v)), \tag{8}$$

where $\hat{X}^v \in \mathbb{R}^{d_v \times n}$ is the obtained $v$th complete view. For self-representation subspace learning with multi-view data, the learned multiple subspaces present a high-order structure, which can be represented by a low-rank tensor. As shown in Figure 2, the multi-view subspace representation exhibits a high-order correlation. Thus, many works establish the following model to capture the high-order correlation [8, 13, 53].

$$\begin{aligned} \min_{\mathcal{Z}} \quad & \mu\|\mathcal{Z}\|_* + \lambda\mathcal{R}(\{E^v\}_{v=1}^V), \\ s.t. \quad & X = XZ^v + E^v, \end{aligned} \tag{9}$$

where $\|\cdot\|_*$ is t-SVD based TNN, $\mu$ and $\lambda$ are balance parameters, $E^v$ denotes the $v$th noise matrix, and $\mathcal{R}$ is the regularizations on $E$.

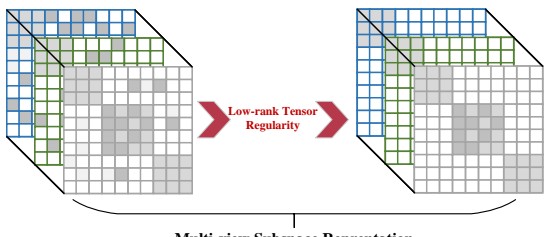

**Figure 2: The visualization of multi-view subspace represen-tation.**

Ordinary TNN treats different singular values equally. This, how-ever, ignores the fact that different singular values hold different contributions. For instance, given an image that is represented as a matrix, the large singular values may contain the salient fea-ture, while small values consist of noise. Therefore, Inspired by the success of weighted TNN and nonconvex low-rank tensor approxi-mation, we designed an enhanced TNN, called Logarithm-$p$ norm. The Logarithm-$p$ norm is formulated as follows.

$$\|Z\|_{log,p} = \sum_i \log(1 + \sigma_i(Z))^p, \tag{10}$$

where $\sigma_i(Z)$ is $i$th singular values of matrix $Z$ with non-descending ordering. For tensor $\mathcal{Z} \in \mathbb{R}^{n_1 \times n_2 \times n_3}$, its Logarithm-$p$ norm is given as

$$\|\mathcal{Z}\|_{log,p} = \sum_{k=1}^{n_3} \|\bar{Z}^k\|_{log,p} = \sum_{k=1}^{n_3} \sum_{i=1}^{\min(n_1,n_2)} \log(1 + \sigma_i(\bar{Z}^k))^p, \tag{11}$$

where $\bar{Z}^k$ denotes the $k$th frontal slice of $\bar{\mathcal{Z}}$ and $\bar{\mathcal{Z}}$ is obtained by conducting Fast Fourier Transformation (FFT) on $\mathcal{Z}$ among third dimension. In the next section, a detailed proof is given to show how the Logarithm-$p$ norm shrinks singular values to preserve the salient feature while eliminating noise.

Finally, the overall objective can be formulated as follows.

$$\min_{\mathcal{Z}, \hat{X}^v, E^v} \quad \sum_{v=1}^{V} (\|X^v + E^v W^v - \hat{X}^v\|_F^2 + \alpha Tr((E^v)^T L_g^v E^v) \\ + \frac{\mu}{2}\|\hat{X}^v - \hat{X}^v Z^v\|_F^2) + \|\mathcal{Z}\|_{log,p}, \tag{12}$$

where $\alpha$ and $\mu$ are balance parameters, $\mathcal{Z} \in \mathbb{R}^{n \times V \times n}$ is a tensor generated by stacking each $Z^v$ and then rotating its dimensionality to $n \times V \times n$. Nevertheless, the complete feature views $\{\hat{X}^v\}_{v=1}^V$ are imputed from the observed information, and the learned sub-spaces are constructed as a rotated tensor to capture the latent high-order correlation. Once the optimal results are computed, off-the-shelf spectral clustering is applied on $S = \frac{1}{V}\sum_{v=1}^V \frac{|Z^v|+|(Z^v)^T|}{2}$ to generate the final partition.

---

**Algorithm 1** Optimization

**Input:** Incomplete multi-view data $\{X^v\}_{v=1}^V$, indicator vectors $\{\mathbf{e}^v\}_{v=1}^V$, and parameters $\{\alpha, \gamma, \mu\}$;
1: **Initialize:**
    Construct index matrices $\{W^v\}_{v=1}^V$ by $\{\mathbf{e}^v\}_{v=1}^V$;
    Construct feature similarity matrix $\{G^v\}_{v=1}^V$ by $k$ nearest neigh-bors algorithm;
    Compute $\{L_g^v\}_{v=1}^V$ by performing $L_g^v = D_g^v - G^v$;
    Initialize $\{J^v\}_{v=1}^V$, $\{Z^v\}_{v=1}^V$, and $\{\bar{X}^v\}_{v=1}^V$ to zero, while $\{E^v\}_{v=1}^V$ are initialized to random matrices.
2: **while** not converging **do**
3:     **for** $v = 1, 2, \cdots, V$ **do**
4:         Update $E^v$ by Equation (20);
5:         Update $\bar{X}^v$ by Equation (16);
6:         Update $Z^v$ by Equation (24);
7:     **end for**
8:     Update $\mathcal{J}$ by Theorem (1);
9: **end while**
10: Compute affinity matrix by

$$S = \frac{1}{V} \sum_{v=1}^{V} \frac{|Z^v| + |(Z^v)^T|}{2};$$

**Output:** The discrete partition which is generated by performing spectral clustering on $S$;

---

## 4 OPTIMIZATION

The Formula (12) is difficult to optimize due to the coupling of variable $\mathcal{Z}$. Thus, an auxiliary variable $\mathcal{J}$ is introduced as follows.

$$\min_{Z^v, \mathcal{J}, \hat{X}^v, E^v} \quad \sum_{v=1}^{V} (\|X^v + E^v W^v - \hat{X}^v\|_F^2 + \alpha Tr((E^v)^T L_g^v E^v) \\ + \frac{\mu}{2}\|\hat{X}^v - \hat{X}^v Z^v\|_F^2) + \|\mathcal{J}\|_{log,p}, \tag{13}$$
$$s.t. \qquad \mathcal{J} = \mathcal{Z}.$$

To solve Formula (13), we devise an iterative optimization algorithm in which each variable is alternately optimized while others are fixed. The overall procedure is summarised in Algorithm 1.

### 4.1 Solution to $\hat{X}^v$

When $E^v$, $Z^v$, and $\mathcal{J}^v$ are fixed, the optimization respects to $\hat{X}^v$ can be written as follows.

$$\min_{\hat{X}^v} \quad \|X^v + E^v W^v - \hat{X}^v\|_F^2 + \frac{\mu}{2}\|\hat{X}^v - \hat{X}^v Z^v\|_F^2. \tag{14}$$

To obtain the local optimal solution of Formula (14), we take the partial derivative concerning $\hat{X}^v$ and set it to zero.

$$2\hat{X}^{(v)} - 2T^v + \mu \hat{X}^{(v)} \check{Z}^v (\check{Z}^v)^T = 0, \tag{15}$$

where $T^v = X^v + E^v W^v$ and $\check{Z}^v = I - Z^v$. The optimal $\hat{X}^{(v)}$ is obtained by

$$\hat{X}^{(v)} = 2T^v (2I - \mu \check{Z}^v (\check{Z}^v)^T)^{-1}. \tag{16}$$

## 4.2 Solution to $E^v$

When $\hat{X}^{(v)}$, $Z^v$, and $\mathcal{J}^v$ are fixed, the objective function of $E^v$ is

$$\min_{E^v} \|X^v + E^v W^v - \hat{X}^v\|_F^2 + \alpha Tr((E^v)^T L_g^v E^v). \quad (17)$$

Since $W^v$ denotes the mapping of absent instances to all instances, we can obtain $W^v(W^v)^T = I$. Due to the unitarily invariant of Frobenius norm, Formula (17) can be transformed into

$$\min_{E^v} \|E^v + (X^v - \hat{X}^v)(W^v)^T\|_F^2 + \alpha Tr((E^v)^T L_g^v E^v). \quad (18)$$

The partial derivative of $E^v$ is set to zero while the following equation is obtained.

$$2E^v + 2(X^v - \hat{X}^v)(W^v)^T + 2L_g^v E^v = 0. \quad (19)$$

The optimal solution to $E^v$ is

$$E^v = (I + L_g^v)^{-1}(\hat{X}^v - X^v)(W^v)^T. \quad (20)$$

## 4.3 Solution to $Z^v$

For the optimization of $Z^v$, the other variables are fixed. The minimization problem can be formulated as follows.

$$\min_{Z^v} \quad \frac{\mu}{2}\|\bar{X}^{(v)} - \bar{X}^{(v)}Z^{(v)}\|_F^2, \\ s.t. \quad J^v = Z^v, \quad (21)$$

where $J^v$ denotes the $v$th frontal slice of tensor $\mathcal{J}$. In order to transfer Formula (21) to an unconstrained optimization function, an augmented Lagrangian function is introduced.

$$\mathcal{L}(Z^v) = \frac{\mu}{2}\|\bar{X}^{(v)} - \bar{X}^{(v)}Z^{(v)}\|_F^2 + \frac{\rho}{2}\|Z^v - J^v + \frac{\Lambda^v}{\rho}\|_F^2, \quad (22)$$

where $\Lambda^v$ is Lagrange multiplier, $\rho$ is a penalty parameter.

Then, the partial derivative concerning $Z^v$ is taken to zeros.

$$\frac{\partial \mathcal{L}(Z^v)}{\partial Z^v} = \mu((\hat{X}^v)^T\hat{X}^v + (\hat{X}^v)^T\hat{X}^v Z^v) + \rho(Z^v - (J^v - \frac{\Lambda^v}{\rho})) \\ = 0. \quad (23)$$

The optimal $Z^v$ is obtained by reformulated Equation (23) as follows.

$$Z^v = (I + \frac{\mu}{\rho}(\hat{X}^v)^T\hat{X}^v)^{-1}((\mu(\hat{X}^v)^T\hat{X}^v - \Lambda^v)/\rho + J^v). \quad (24)$$

## 4.4 Solution to $\mathcal{J}$

To update $\mathcal{J}$, the following theorem is introduced.

**Theorem 1** Given a minimization problem as

$$\min_{\mathcal{J}} \|\mathcal{J}\|_{log,p} + \frac{\rho}{2}\|\mathcal{J} - \mathcal{P}\|_F^2, \quad (25)$$

where $\mathcal{J}$ and $\mathcal{P}$ are both tensor with size $m \times V \times n$. The optimal $\mathcal{J}$ can be obtained by the following steps.

(1) Transform $\mathcal{P}$ into $\hat{\mathcal{P}}$ by performing FFT on $\mathcal{P}$ among third dimension.

(2) Updated $\hat{\mathcal{J}}^i$ can be obtained by $\hat{\mathcal{J}}^i = U^i S_{\nabla\phi/\rho}(\Sigma^i)(Y^i)^T$, where $\hat{\mathcal{P}}^i = U^i \Sigma^i Y^i$, $\nabla\phi$ denotes the derivative of $\log(1+x)^p$, and the formulation of $S_{\nabla\phi/\rho}(\Sigma^i)$ is

$$S_{\nabla\phi/\rho}(\Sigma^i) = diag(\max\{\Sigma_{v,v}^i - \frac{\nabla\phi(\Sigma_{v,v}^i)}{\rho}, 0\}_{v=1}^V)$$

(3) Then, stacking each $\hat{\mathcal{J}}^i$ to $\hat{\mathcal{J}}$. The updated $\mathcal{J}$ can be computed by performing inverse FFT on $\hat{\mathcal{J}}$ among third dimension.

PROOF. Inherited from Equation (11), the Formula (25) can be divided into $n$ subproblems, while the $i$th subproblem can be written as

$$\min_{\hat{J}^i} \|\hat{J}^i\|_{log,p} + \frac{\rho}{2}\|\hat{J}^i - \hat{\mathcal{P}}^i\|_F^2. \quad (26)$$

According to the non-ascending order of singular values and the monotonicity of Equation (11), Formula (26) can be relaxed as the following form.

$$\min_{\hat{J}^i} \sum_{v=1}^V (\phi(\sigma_v) + \nabla\phi(\sigma_v)(\sigma_v(\hat{J}^i) - \sigma_v)) + \frac{\rho}{2}\|\hat{J}^i - \hat{\mathcal{P}}^i\|_F^2, \quad (27)$$

where $\sigma_v$ denotes the $v$th singular value of current $\hat{J}^i$, $\sigma_v(\hat{J}^i)$ represents the $v$th singular value of updated $\hat{J}^i$. All singular values are non-decreasing. Since $\phi(\sigma_v)$ and $\sigma_v$ are constants, we rewrite the above problem as

$$\min_{\hat{J}^i} \sum_{v=1}^V \nabla\phi(\sigma_v)\sigma_v(\hat{J}^i) + \frac{\rho}{2}\|\hat{J}^i - \hat{\mathcal{P}}^i\|_F^2. \quad (28)$$

The optimal solution of $\hat{J}^v$ can be obtained by generalized weighted singular value thresholding [31], which can be formulated as

$$\hat{J}^i = U^i S_{\nabla\phi/\rho}(\Sigma^i)(Y^i)^T, \quad (29)$$

where $\nabla\phi_i = p/(1 + \Sigma^i)$, meaning the shrinkage weights are governed by $\Sigma^i$. Thus, we can intuitively obtain that $\nabla\phi_i$ is non-decreasing due to $\Sigma^i$ being non-increasing, which leads to large singular values being preserved. $\square$

By fixing $\hat{X}^{(v)}$, $E^v$, and $Z^v$, the optimization of $\mathcal{J}$ can be formulated as

$$\min_{\mathcal{J}} \|\mathcal{J}\|_{log,p} + \frac{\rho}{2}\|\mathcal{J} - (\mathcal{Z} + \frac{\mathcal{T}}{\rho})\|_F^2, \quad (30)$$

where $\mathcal{T} \in \mathbb{R}^{n \times V \times n}$ is a 3-order tensor constructed by stacking each $\Lambda^v$ and rotating its dimension into $n \times V \times n$. The updated rule of Formula (30) is shown in Theorem (1).

## 4.5 Solution to Multiplier and Penalty Parameter

The Lagrange multiplier and penalty parameter can be updated by

$$\Lambda^v = \Lambda^v + \rho(Z^v - J^v), \quad (31)$$

$$\rho = \min(\beta * \rho, \rho_{max}), \quad (32)$$

where $\beta$ denotes the growth rate and is set to 1.5. $\rho_{max}$ is maximum value of penalty parameter [2].

## 4.6 Convergence Condition

Similar to [4, 29], the convergence condition is prescribed as

$$\frac{|obj_{t-1} - obj_t|}{|obj_t|} < \epsilon, \quad (33)$$

where $obj_t$ represents the value of objective function at the $t$th iteration, and $\epsilon$ is a pre-defined parameter.

**Table 1: Incompete multiview datasets in experiments.**

| Datasets | Size | Clusters | Views | Dimensionality |
|---|---|---|---|---|
| 3source | 169 | 6 | 3 | 3560/3631/3068 |
| MSRC v1 | 210 | 7 | 6 | $1302/48/\cdots/256/210$ |
| Yale | 165 | 15 | 3 | 4096/3304/6750 |
| ORL | 400 | 40 | 3 | 4096/3304/6750 |
| BBC | 685 | 5 | 4 | 4659/4633/4665/4684 |
| BBCSport | 544 | 5 | 2 | 3283/3183 |
| Caltech7 | 1474 | 7 | 6 | 48/40/.../512/928 |
| Leaves | 1600 | 100 | 3 | 64/64/64 |

## 4.7 Convergence Analysis

Algorithm 1 decomposes the optimization problem into four sub-problems. Each subproblem enjoys a closed-form solution. Without loss of generality, the value of the overall objective function decreases with the solution of each variable. Since the non-negative property of Formula (13), the designed iterative alternation algorithm ensures the proposed method converges to a local optimum solution.

## 4.8 Computational Complexity Analysis

Following, we analyze the computational complexity in detail. The main complexity of Algorithm 1 depends on the iteration procedure. Firstly, the computation of $E^v$ concerning the matrix inversion. However, the inverse of $(I + L_g^v)$ can be computed in advance. Therefore, the computational complexity of updating $E^v$ is $O(d_v^2 n + d_v n n_v^m)$ resulting from matrix multiplication. The optimization of $\bar{X}^v$, involves computing the inverse matrix, with a complexity of $O(n^3)$. For updating $Z^v$, the main complexity is $O(n^3)$, which comes from computing inverse matrix. Here $\mathcal{J} \in \mathbb{R}^{n \times V \times n}$, the updating $\mathcal{J}$ needs FFT, singular value decomposition, and inverse FFT. The computational complexity of FFT and IFFT is $O(Vn^2 log(n))$, and the step of singular value decomposition costs $O(n^2 V^2)$. The total complexity of Formula (30) is $O(Vn^2 log(n) + n^2 V^2)$. Thus, the overall computational complexity of the proposed method is $O(t(d_v^2 n + n^3))$, where $t$ is the number of iterations. The computational complexity of Algorithm 1 is dependent on the cubic of instances and the dimension of the feature, while the number of iterations also affects the computation consumption.

## 5 EXPERIMENT

In this section, to verify the performance of the proposed method, several widely-used datasets are chosen. Meanwhile, eleven state-of-the-art IMVC algorithms are adopted as comparison methods. In addition, model analyses, including ablation study, parameters analysis, and convergence analysis, are performed to explore the proposed model. At the very beginning, the experiment setup is introduced.

## 5.1 Experiment Setup

*5.1.1 Datasets Description.* Nine multi-view datasets are adopted, including 3source, MSRC v1, Yale, ORL, BBC, BBCSport, Caltech7, Leaves, and UCI digits. Specifically, the detailed information of all datasets is summarised in Table 1.

*5.1.2 Incomplete Multi-view Datasets Construction.* Following the principle that each sample appears in at least one view, incomplete multi-view datasets are generated with missing ratios at 0.2 intervals from 0.1 to 0.9. Particularly, for an incomplete sample, it is randomly absent in some views, however, present in at least one view.

*5.1.3 Compared Methods.* Eleven state-of-the-art models are chosen, including APMC [16], BGIMVSC [38], CBG [44], DAIMC [17], FIMVC [30], FLSD [49], HCP [25], IMSR [29], PGP [27], PIC [42], and SIMVC [50].

For compared methods, parameters are tuned in the range advised in the literature to report the best results. For the proposed method, three parameters, including $\alpha$, $\gamma$, and $\mu$ are adjusted to $[10^{-4}, 10^{-3}, 10^{-2}, 10^{-1}, 10^0, 10^1, 10^2, 10^3, 10^4]$, $[5, 10, 15, 20, 50, 100, 150, 200, 300]$, and $[10^{-4}, 10^{-3}, 10^{-2}, 10^{-1}, 10^0, 10^1, 10^2, 10^3, 10^4]$, respectively. We run each method 10 cycles to calculate average performance and standard deviation for reporting.

*5.1.4 Evaluation Metrics and Experiment Environment.* To assess the performance of clustering, three metrics, Accuracy (ACC), Normalized Mutual Information (NMI), and Purity are adopted.

Experiments are conducted on MATLAB 2022b (64-bit) with Intel Xeon Gold 6248R CPU, 256G RAM.

## 5.2 Compared Results

Table 2 reports the experiment results on all datasets with 10% samples are incomplete. The bold and underlined indicate the best and suboptimal results, respectively, while '−' denotes the unavailable results due to program error. Figure 3 shows the results on all datasets with different missing rates. To visualize the running time of different methods, we convert real runtime to relative Logarithm runtime, which is reported in Figure 4. These provide the following observations.

(1) The proposed method achieves superior performance on all datasets with 10% incomplete samples. Meanwhile, PGP surpasses other methods except ELRSSL. For instance, compared to the suboptimal approach, ELRSSL has 14.14%, 1.1%, 6.87%, 6.19%, 8.45%, 19.82, 3.48%, 9.64%, and 13.92% improvement on terms of ACC. The excellent performance demonstrates the critical role of missing instances completion and low-rank tensor regularity.

(2) Compared to IMSR, which is a pioneering work in self-representation learning for IMVC, the proposed method is superior. The difference between IMSR and our ELRSSL is that ELRSSL infers samples from the perspective of feature and efficiently explores high-order information among multiple views, resulting in a more accurate representation of subspaces. Compared to HCP, which controls the tensor tubal rank by tensor factorization, the proposed method offers better clustering results. It demonstrates the effectiveness of the proposed Logarithm-$p$ norm.

(3) As shown in Figure 3, the proposed method is superior on several datasets, such as ORL, Yale, Leaves, and UCI digits. However, ELRSSL shows great fluctuations in performance on BBCSport and Caltech7 datasets. We conjecture that this is because the ORL, Yale, Leaves, and UCI digits datasets

**Table 2: Comparison results on nine datasets with missing rate** 10%.

| Metrics | Datasets | APMC[16] | BGIMVSC[38] | CBG[44] | DAIMC[17] | FIMVC[30] | FLSD[49] | HCP [25] | IMSR[29] | PGP [27] | PIC [42] | SIMVC[50] | ELRSSL |
|---|---|---|---|---|---|---|---|---|---|---|---|---|---|
| **ACC** | 3source | $68.40_{0.31}$ | $69.23_{0.00}$ | $54.32_{0.78}$ | $56.39_{4.79}$ | $66.27_{0.00}$ | $67.46_{0.00}$ | $71.36_{1.68}$ | $65.33_{0.31}$ | $82.31_{0.44}$ | $64.08_{0.40}$ | $56.92_{0.37}$ | $\mathbf{96.45_{0.00}}$ |
| | BBCsports | $97.43_{0.00}$ | $97.43_{0.00}$ | $89.15_{0.00}$ | $79.23_{8.98}$ | $90.99_{0.00}$ | $96.14_{0.00}$ | $96.14_{0.00}$ | $95.37_{0.08}$ | $97.98_{0.00}$ | $97.79_{0.00}$ | $83.27_{0.00}$ | $\mathbf{99.08_{0.00}}$ |
| | BBC | – | $85.11_{0.00}$ | $63.14_{0.08}$ | $61.87_{4.13}$ | $80.04_{0.07}$ | $84.53_{0.00}$ | $88.18_{0.00}$ | $91.68_{0.00}$ | $92.55_{0.00}$ | $91.62_{0.10}$ | $65.81_{0.59}$ | $\mathbf{99.42_{0.00}}$ |
| | MSRC v1 | – | $65.24_{0.00}$ | $70.48_{0.00}$ | $74.76_{5.26}$ | $86.19_{0.78}$ | $73.05_{0.33}$ | $90.90_{0.15}$ | $85.05_{0.60}$ | $92.86_{0.03}$ | $80.48_{0.00}$ | $64.62_{0.64}$ | $\mathbf{99.05_{0.00}}$ |
| | ORL | $74.87_{1.77}$ | $74.00_{0.00}$ | $63.95_{0.61}$ | $67.67_{3.25}$ | $75.92_{2.12}$ | $61.35_{2.54}$ | $81.17_{1.83}$ | $79.85_{1.31}$ | $80.05_{0.64}$ | $76.80_{1.76}$ | $64.08_{1.85}$ | $\mathbf{89.62_{1.49}}$ |
| | Yale | $63.21_{3.41}$ | $60.61_{0.00}$ | $57.45_{0.48}$ | $50.00_{4.53}$ | $71.03_{1.02}$ | $53.45_{1.95}$ | $73.76_{0.57}$ | $73.09_{1.77}$ | $72.12_{0.00}$ | $70.12_{1.87}$ | $57.09_{3.27}$ | $\mathbf{93.58_{4.82}}$ |
| | Leaves | $84.20_{0.74}$ | $87.88_{0.00}$ | $29.94_{0.39}$ | $62.57_{2.71}$ | $63.21_{1.61}$ | $60.06_{1.78}$ | $76.94_{1.72}$ | $76.08_{1.80}$ | $88.63_{1.40}$ | $68.88_{2.24}$ | $52.57_{1.23}$ | $\mathbf{92.11_{1.34}}$ |
| | UCI digits | $76.80_{0.00}$ | $78.10_{0.00}$ | $26.71_{0.14}$ | $74.20_{9.08}$ | $86.05_{0.11}$ | $88.48_{0.18}$ | $85.84_{0.04}$ | $89.61_{0.02}$ | $75.92_{0.12}$ | $71.89_{0.12}$ | $67.49_{4.05}$ | $\mathbf{99.25_{0.00}}$ |
| | Caltech101-7 | – | $57.80_{0.00}$ | $48.44_{0.00}$ | $44.80_{5.57}$ | $51.66_{0.04}$ | $58.11_{0.12}$ | $57.33_{0.31}$ | $61.59_{0.17}$ | $66.55_{0.11}$ | $64.06_{0.67}$ | $54.71_{1.59}$ | $\mathbf{80.47_{0.18}}$ |
| **NMI** | 3source | $68.70_{0.61}$ | $60.09_{0.00}$ | $52.28_{0.39}$ | $55.95_{3.16}$ | $67.18_{0.00}$ | $41.22_{0.00}$ | $61.82_{0.77}$ | $59.08_{0.48}$ | $67.98_{0.62}$ | $65.02_{1.22}$ | $49.04_{0.59}$ | $\mathbf{90.47_{0.00}}$ |
| | BBCsports | $90.98_{0.00}$ | $91.43_{0.00}$ | $78.52_{0.00}$ | $73.61_{3.23}$ | $83.15_{0.00}$ | $87.38_{0.00}$ | $87.54_{0.00}$ | $86.06_{0.17}$ | $92.48_{0.00}$ | $91.96_{0.00}$ | $66.09_{0.00}$ | $\mathbf{96.89_{0.00}}$ |
| | BBC | – | $69.46_{0.00}$ | $55.94_{0.14}$ | $56.45_{3.03}$ | $62.18_{0.07}$ | $67.81_{0.00}$ | $70.71_{0.00}$ | $77.31_{0.00}$ | $79.11_{0.00}$ | $77.55_{0.20}$ | $43.79_{0.21}$ | $\mathbf{97.89_{0.00}}$ |
| | MSRC v1 | – | $61.61_{0.00}$ | $60.34_{0.00}$ | $66.32_{2.27}$ | $74.84_{0.96}$ | $63.89_{0.55}$ | $81.92_{0.42}$ | $74.42_{0.88}$ | $85.59_{0.00}$ | $73.07_{0.00}$ | $54.06_{0.62}$ | $\mathbf{97.84_{0.00}}$ |
| | ORL | $86.93_{0.98}$ | $84.05_{0.00}$ | $80.82_{0.43}$ | $83.00_{1.00}$ | $87.31_{1.01}$ | $77.45_{1.32}$ | $89.25_{0.79}$ | $89.40_{0.60}$ | $89.71_{0.22}$ | $88.07_{0.66}$ | $78.30_{0.98}$ | $\mathbf{96.31_{0.68}}$ |
| | Yale | $68.03_{1.56}$ | $66.09_{0.00}$ | $62.10_{0.58}$ | $57.20_{3.51}$ | $71.34_{0.67}$ | $58.04_{1.78}$ | $74.63_{0.67}$ | $73.03_{0.65}$ | $73.96_{0.00}$ | $72.22_{0.92}$ | $57.71_{1.32}$ | $\mathbf{94.69_{3.30}}$ |
| | Leaves | $92.99_{0.31}$ | $92.35_{0.00}$ | $58.43_{0.65}$ | $81.42_{1.24}$ | $80.50_{0.57}$ | $80.14_{0.95}$ | $87.72_{0.70}$ | $88.08_{0.53}$ | $94.59_{0.38}$ | $82.93_{1.04}$ | $73.47_{0.59}$ | $\mathbf{97.44_{0.30}}$ |
| | UCI digits | $76.94_{0.00}$ | $78.32_{0.00}$ | $29.27_{0.12}$ | $70.25_{3.80}$ | $75.93_{0.11}$ | $79.92_{0.27}$ | $75.25_{0.07}$ | $80.80_{0.06}$ | $77.36_{0.08}$ | $73.24_{0.08}$ | $58.67_{2.51}$ | $\mathbf{98.05_{0.00}}$ |
| | Caltech101-7 | – | $45.18_{0.00}$ | $38.21_{0.00}$ | $41.36_{1.49}$ | $36.82_{0.11}$ | $24.44_{0.17}$ | $45.78_{0.13}$ | $51.38_{0.16}$ | $55.87_{0.86}$ | $48.97_{0.64}$ | $01.26_{2.70}$ | $\mathbf{63.99_{0.25}}$ |
| **Purity** | 3source | $76.75_{1.05}$ | $76.92_{0.00}$ | $72.78_{0.00}$ | $72.54_{3.03}$ | $82.25_{0.00}$ | $69.23_{0.00}$ | $76.09_{0.64}$ | $76.57_{0.31}$ | $82.31_{0.44}$ | $80.00_{0.54}$ | $72.13_{0.19}$ | $\mathbf{96.45_{0.00}}$ |
| | BBCsports | $97.43_{0.00}$ | $97.43_{0.00}$ | $89.15_{0.00}$ | $83.82_{1.87}$ | $90.99_{0.00}$ | $96.14_{0.00}$ | $96.14_{0.00}$ | $95.37_{0.08}$ | $97.98_{0.00}$ | $97.79_{0.00}$ | $83.27_{0.00}$ | $\mathbf{99.08_{0.00}}$ |
| | BBC | – | $85.11_{0.00}$ | $72.34_{0.08}$ | $75.08_{2.55}$ | $80.04_{0.07}$ | $84.53_{0.00}$ | $88.18_{0.00}$ | $91.68_{0.00}$ | $92.55_{0.00}$ | $91.62_{0.10}$ | $66.10_{0.90}$ | $\mathbf{99.42_{0.00}}$ |
| | MSRC v1 | – | $67.14_{0.00}$ | $71.43_{0.00}$ | $75.95_{3.97}$ | $86.19_{0.78}$ | $73.48_{0.23}$ | $90.90_{0.15}$ | $85.05_{0.60}$ | $92.86_{0.00}$ | $80.48_{0.00}$ | $65.10_{0.64}$ | $\mathbf{99.05_{0.00}}$ |
| | ORL | $79.45_{1.41}$ | $75.75_{0.00}$ | $69.12_{0.56}$ | $71.40_{2.75}$ | $78.67_{1.65}$ | $63.60_{2.18}$ | $82.67_{1.80}$ | $82.12_{0.87}$ | $82.05_{0.45}$ | $79.25_{1.51}$ | $66.60_{1.38}$ | $\mathbf{92.02_{1.02}}$ |
| | Yale | $73.82_{1.24}$ | $61.21_{0.00}$ | $58.06_{0.48}$ | $52.12_{3.99}$ | $71.03_{1.02}$ | $53.94_{1.62}$ | $73.76_{0.57}$ | $73.27_{1.63}$ | $72.12_{0.00}$ | $70.18_{1.69}$ | $57.64_{3.39}$ | $\mathbf{93.58_{4.82}}$ |
| | Leaves | $90.76_{0.55}$ | $89.00_{0.00}$ | $32.78_{0.39}$ | $65.96_{2.61}$ | $65.89_{1.17}$ | $61.59_{1.65}$ | $78.94_{1.32}$ | $78.27_{1.44}$ | $90.03_{0.94}$ | $71.41_{1.97}$ | $54.91_{1.07}$ | $\mathbf{93.64_{0.93}}$ |
| | UCI digits | $79.88_{0.04}$ | $79.45_{0.00}$ | $27.37_{0.13}$ | $76.11_{7.23}$ | $86.05_{0.11}$ | $88.48_{0.18}$ | $85.84_{0.04}$ | $89.61_{0.02}$ | $75.92_{0.12}$ | $72.17_{0.08}$ | $67.81_{3.78}$ | $\mathbf{99.25_{0.00}}$ |
| | Caltech101-7 | – | $83.04_{0.00}$ | $81.48_{0.00}$ | $83.36_{1.76}$ | $80.70_{0.11}$ | $73.70_{0.14}$ | $84.88_{0.02}$ | $86.53_{0.07}$ | $87.88_{0.58}$ | $84.72_{0.34}$ | $55.06_{1.63}$ | $\mathbf{91.40_{0.04}}$ |

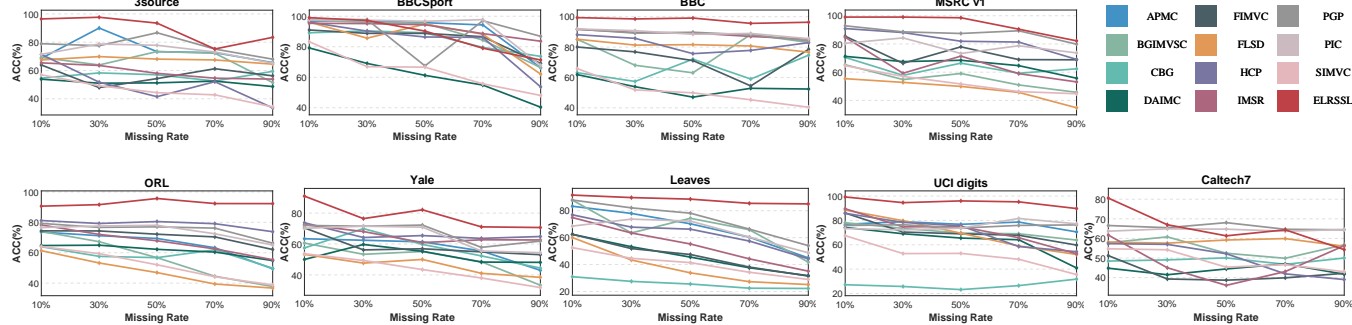

**Figure 3: Comparison results on nine datasets with different missing ratios.**

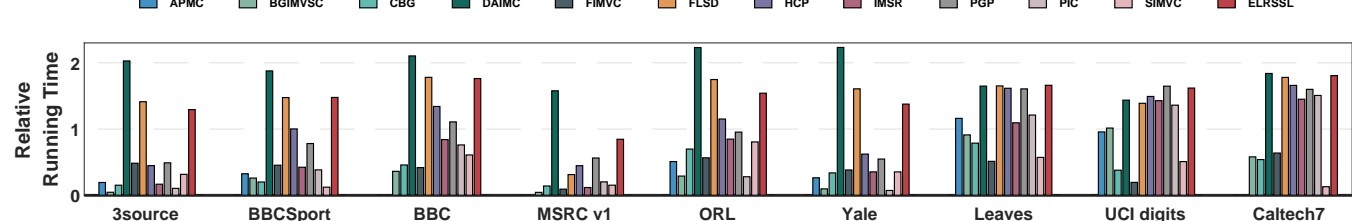

**Figure 4: Time report of different methods on nine incomplete multi-view datasets.**

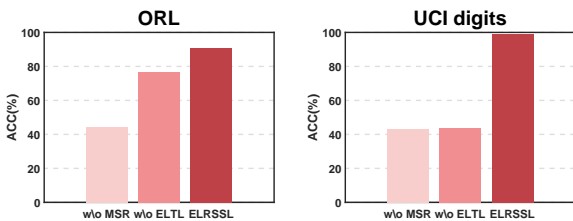

Figure 5: The ablation study.

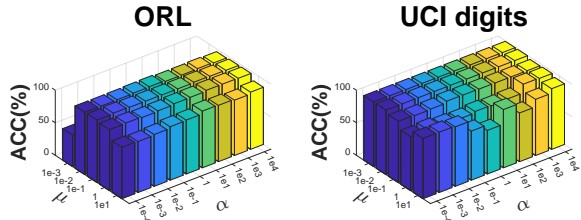

Figure 6: The sensitivity analysis of parameters $\alpha$ and $\mu$.

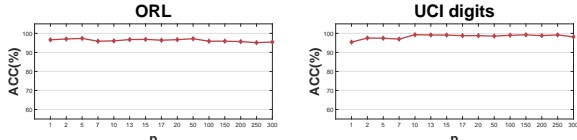

Figure 7: The sensitivity analysis of parameter $p$.

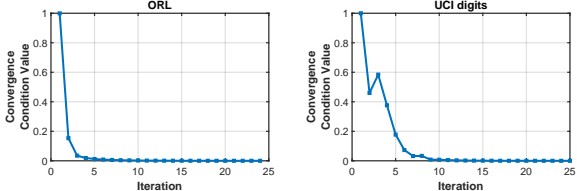

Figure 8: The convergence analysis on two datasets.

enjoy well-structured subspace, e.g. the face datasets (ORL, Yale) hold low-rank properties and are more suitable for self-representation subspace learning [1, 58]. Overall, the proposed ELRSSL is superior on most datasets with various missing ratios, demonstrating its superiority in IMVC task.

(4) Compared to anchor-based methods, ELRSSL requires more time consumption but has better performance. Furthermore, the proposed method is more time-consuming than the sub-optimal method PGP, due to the computation of matrix inversion and tensor singular value decomposition. In general, the extra time consumption is worthwhile as the proposed method achieves remarkable clustering performance.

## 5.3 Model Analysis

*5.3.1 Ablation Study.* An ablation study is conducted to examine the validity of the proposed method. Based on the architecture of ELRSSL, there are two crucial modules, including missing sample recovery and low-rank tensor learning. Therefore, the following two variants are designed.

(1) w\o MSR (without Missing Sample Recovery): This variant removes the $\alpha Tr((E^v)^T L_g^v E^v)$ ($\alpha$ is set to 0) and retains enhanced low-rank tensor learning.

(2) w\o ELTL (without Enhanced Low-rank Tensor Learning) This variant replaces the Logarithm-$p$ norm with the ordinary TNN, which is formulated as $\|\mathcal{J}\|_* = \sum_k \sum_i \sigma_i(\bar{J}^k)$ while missing sample recovery is retained.

Figure 5 shows the results of the ablation study. The experiment results verify the importance of missing sample recovery by leveraging local structure and capturing high-order correlation among multiple views by enhanced low-rank tensor regularity, both of which contribute to clustering.

*5.3.2 Parameter Sensitivity Analysis.* There are three main hyper-parameters for the proposed method, including $\alpha$, $\gamma$, and $p$. Particularly, the number of neighbors for $k$-nearest neighbor in Equation (6) is fixed to 7 due to its hardness in selecting the optimal value. The tuning range of these parameters is given in the section 5.1. As shown in Figure 6 and Figure 7, the proposed method enjoys stable performance.

*5.3.3 Convergence Analysis.* As shown in Figure 8, the objective value decreases in each iteration, which proves the proposed method can converge rapidly.

## 6 CONCLUSION

In this paper, an effective self-representation subspace learning method is devised for IMVC. In particular, a local structure in each view is introduced to recover the missing samples. Meanwhile, we designed an enhanced low-rank tensor regularity to efficiently capture the high-order correlation by assigning different weights to different singular values. Extensive experiments demonstrate the superiority of the proposed method and the improvement compared with previous self-representation subspace learning approaches. We plan to explore an efficient mechanism without precision reduction for self-representation on the IMVC task in future.

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
