# OpenReview forum: "Enhanced Tensorial Self-representation Subspace Learning for Incomplete Multi-view Clustering"
_acmmm.org/ACMMM/2024/Conference — MM2024 Poster_

### Official Review · Reviewer_eJqm · 2024-05-20

**Rating:** 5
**Confidence:** 4

**Summary:**

The authors introduce ELRSSL, a new method to group incomplete multi-view samples into corresponding clusters. ELRSSL works by (1) recovering the missing instance by restricting similar features which ensures the inferred entries are meaningful, and (2) devising an enhanced low-rank tensor learning strategy to explore the high-order correlation across multiple views. Comparative analysis against state-of-the-art methods underscores the effectiveness of ELRSSL, showcasing the potential of self-representation learning in incomplete multi-view clustering.

**Strengths:**

1. The authors recognize and address drawbacks in prior self-representation learning based IMVC approaches, which are (1) redundant information resulting from incomplete data inference and (2) overlooking salient features in the self-representation subspace.

2. The evaluation performed is comprehensive and the performances of the proposed method are impressive.

3. The comparison results and the ablation study show that differential treatment of different singular values in the learned tensor is beneficial for clustering, reinforcing the author's assertions in Section 1.

4. The paper's organization facilitates ease of comprehension.

**Limitations:**

These are not major weaknesses, but suggestions to strengthen the results:

1. There are inconsistencies in the notations. In Eq. (5), I think "$E{j :}^v$" is not what the authors want to represent, perhaps "$E_{j :}^v$". Authors are required some careful revisions.

2. I would like the authors to further discuss the performances of IMVC methods if all samples are visible in only one view.

3. The authors claim that the "-" in Table 2 is due to a program error, and I would like the authors to explain the cause of the error.

4. In Figure 4, it is not clear what is relative running time in this paper.

5. It is not clear how the number of neighbors in constructing guidance matrix G will individually impact the model's performance.

6. The performances of CBG on the UCI digits and Leaves datasets are not promising. Please provide the explanation and analyses.

7. Have you considered explaining the intrinsic structure of high-order correlation? In other words, why this structure is formed?

**Suitability:**

3

---

### Official Review · Reviewer_g6B6 · 2024-05-22

**Rating:** 5
**Confidence:** 3

**Summary:**

This paper proposes an incomplete multi-view clustering method called ELRSSL. Specifically, the authors employ self-representation learning to explore relationships among samples. They introduce a strategy for missing sample recovery and enhanced low-rank tensor regularization. Through leveraging local structure and high-order correlations across multiple views, ELRSSL achieves precise subspace representation in IMVC. Extensive experiments on various multi-view datasets demonstrate the effectiveness of the proposed method.

**Strengths:**

1. This paper is well-organized and easy to follow.
2. By incorporating local structure and high-order correlations across multiple views, ELRSSL achieves precise subspace representation, demonstrating its effectiveness in handling complex multiview datasets.
3. This paper proposes a novel self-adaptive weight strategy to retain salient features while eliminating noise.

**Limitations:**

1. Clarity: The flowchart shown in the paper is not clear enough and the meaning of each matrix in Figure 1 is ambiguous. Authors are suggested to add more description.
2. Motivation:
(a) While the concept that the large singular values may contain the salient feature and small values consist of noise is reasonable, the motivation of the proposed Logarithm-p norm is not convincing enough. Key points to explore include: Why can Logarithm-p norm better retain important features than normal TNN? What is the essential difference between the Logarithm-p norm and nonconvex low-rank tensor approximation? What are the physical significance and distribution trends of the proposed Logarithm-p norm?
(b) In this paper, authors declare that the strategy of inferring missing samples used in previous works results in redundant information. To highlight the advantages of the strategy used in this paper, authors are encouraged to provide a detailed comparison of the two strategies.
(c) The work in [Zhang et al., 2023] provides a generalized nonconvex tensor low-rank regularization to alleviate the biased approximation in the self-representation learning framework. A theoretical analysis of the difference between [Zhang et al., 2023] and the proposed method can enhance the comprehensiveness of the article.
[Zhang et al., 2023] Enhanced Tensor Low-Rank and Sparse Representation Recovery for Incomplete Multi-View Clustering. In AAAI.

3. Experiments:
(a) It would be beneficial to categorize comparative methods to clarify the basic framework of the models in reporting experiment results.
(b) Additional Investigation: Explore the impact of different cluster numbers on various metrics.

**Suitability:**

3

---

### Official Review · Reviewer_m8bS · 2024-05-24

**Rating:** 5
**Confidence:** 4

**Summary:**

This paper proposes a novel self-representation learning method for incomplete multi-View clustering. The core motivation is to learn an exact self-representation for IMVC. To compensate for missing information, the authors propose constraining similar features to ensure the recovered data is meaningful, alongside devising an enhanced tensor norm to accurately delineate cross-view relationships. Experiments on various real-world datasets demonstrate the effectiveness of the proposed method.

**Strengths:**

(1) The paper is well-written. The working mechanism and motivation of each module are clearly explained.
(2) The proposed method achieves superior results on numerous benchmarks.
(3) Ablation studies are conducted to demonstrate the effectiveness of the proposed components, while comparisons of running times provide insight into the method's efficiency relative to others.

**Limitations:**

(1) The authors exclusively focus on self-representation subspace learning, overlooking previous works addressing noise in representations. It would be beneficial for the authors to expound upon their rationale for this omission.
(2) The number of clusters is directly specified in experiments. However, it is unknown in the real scenario. Further discussion of the impact of cluster counts is necessary.
(3) Grammatical errors, such as "Guo et al. [16] captures" instead of "Guo et al. [16] capture," should be rectified.
(4) Some papers are incorrectly cited, such as: "Xiaojun Chen, Weijun Hong, Feiping Nie, Dan He, Min Yang, and Joshua Zhexue Huang. 2018. Spectral Clustering of Large-scale Data by Directly Solving Normalized Cut. In Proceedings of the 24th ACM SIGKDD International Conference on Knowledge Discovery & Data Mining, KDD 2018, London, UK, August 19-23, 2018 1206–1215"

**Suitability:**

3

---

### Official Review · Reviewer_49iG · 2024-05-25

**Rating:** 4
**Confidence:** 3

**Summary:**

This paper introduces an Enhanced Tensorial Self-representation Subspace Learning (ELRSSL) method for Incomplete Multi-view Clustering (IMVC). The proposed method addresses the challenge of incomplete data in multi-view clustering by recovering missing samples through local structure inference and leveraging enhanced low-rank tensor regularization.

**Strengths:**

- **Clarity and Readability**: The paper is well-written and easy to understand, making complex concepts accessible to a broader audience.
- **Innovative Approach**: The method shows certain novelty, particularly in integrating local structure inference with enhanced low-rank tensor regularization for missing sample recovery.
- **Extensive Experiments**: The superiority of the proposed method is demonstrated with extensive experiments across multiple datasets, highlighting its effectiveness and robustness.
- **Enhanced Low-rank Tensor Norm**: The introduction of the Logarithm-𝑝 norm for low-rank tensor regularization is a significant contribution, as it allows for better preservation of important features and elimination of noise.

**Limitations:**

* Insufficient Theoretical Justification: The paper would benefit from a more detailed theoretical justification. Especially, it would be better if it could be proved theoretically that the proposed enhanced TNN outperforms TNN .
* Typographical and Writing Errors: There are several minor typographical and grammatical errors throughout the paper. Examples include:"Incompete" should be corrected to "Incomplete" (Table 1 title); The symbol in Eq. 5, $Ej,:$ should be $E_{j,:}$.

* Despite that the work shows several contributions with the enhanced TNN, the overall novelty seems still limited, since several design such as Laplacian Matrix based regularization is has been used by several works.

**Suitability:**

3

---

### Meta-Review · Area_Chair_rqBP · 2024-07-01

**Recommendation:** Accept (Poster)
**Confidence:** 5

**Metareview:**

In this paper, for the challenging issue of incomplete multi-view clustering, the authors proposed a novel Tensorial Self-representation Subspace Learning method. All reviewers unanimously recognized the innovation and quality of this paper and gave an acceptance opinion. The final scores of the paper are Borderline Accept, Weak Accept, and two Accepts after rebuttal. Based on the recommendations of reviewers and the quality of the paper, the final recommendation to the paper is acceptance.